# Melatonin Treatment of Strawberry Fruit during Storage Extends Its Post-Harvest Quality and Reduces Infection Caused by *Botrytis cinerea*

**DOI:** 10.3390/foods12071445

**Published:** 2023-03-29

**Authors:** Surassawadee Promyou, Yenjit Raruang, Zhi-Yuan Chen

**Affiliations:** 1Department of Agriculture and Resources, Faculty of Natural Resources and Agro-Industry, Chalermphrakiat Sakon Nakhon Province Campus, Kasetsart University, Sakon Nakhon 47000, Thailand; 2Department of Plant Pathology and Crop Physiology, Louisiana State University Agricultural Center, Baton Rouge, LA 70803, USA; yraruang@agcenter.lsu.edu (Y.R.); zchen@agcenter.lsu.edu (Z.-Y.C.)

**Keywords:** melatonin, disease resistance, strawberry, gray mold, antioxidant defense system

## Abstract

Gray mold is a main disease of strawberry fruit (*Fragaria* × *xananassa* cv. Camino Real) caused by *Botrytis cinerea*, which leads to marketable value losses in the supply chain. The purpose of this study was to investigate the effects of exogenous melatonin (MT) on the physicochemical quality, antioxidant defense system, and disease resistance of strawberry fruit to *B. cinerea* infection. The results revealed that strawberry fruit immersed in 100 µM MT for 15 min effectively maintained its brightness and delayed the change in fruit color. MT also maintained the level of titratable acidity and slowed down the increase of total soluble solids in strawberry fruit. Moreover, strawberries immersed in MT maintained a fresh weight and fruit firmness, as well as reduced *B. cinerea* infection when compared to the untreated control fruit and fruit treated with 5% NaOCl. In addition, MT increased the accumulation of DPPH scavenging capacity and the activity of antioxidant enzymes (SOD, POD, and APX) with the exception of CAT. The same effect was also observed in strawberry fruit after immersion in MT and followed by *B. cinerea* inoculation. These findings demonstrated that exogenous MT could effectively maintain the postharvest quality of strawberries, even when the fruit was inoculated with *B. cinerea*.

## 1. Introduction

Strawberries (*Fragaria × ananassa* cv. Camino Real) are a commercial non-climacteric fruit that belong to the Rosales family and are well known as a rich source of different types of nutrients. They contain many beneficial health compounds, such as high levels of antioxidants (polyphenolics, flavonoids, vitamin C, and anthocyanins), folate, potassium, and fiber [1]. Strawberries are produced in many countries. In 2020, the United States produced about 4510 thousand tons of strawberries for the market. Recently, strawberries have risen to become the third most important agricultural commodity, in terms of fruit, in the U.S. after grapes and oranges. Interestingly, strawberries are the fifth most consumed fresh fruit by weight, after bananas, apples, oranges, and grapes [2]. The most serious postharvest problem that results in fruit deterioration during storage and that affects strawberry fruit quality is the gray mold disease caused by *Botrytis cinerea* [3].The initial symptoms of the infected strawberry fruit are light brown lesions and a loss of fruit firmness, which cover the entire fruit surface within 48 h. Gray mold can result in losses ranging from 25–35%, or approximately USD 10–100 million per year. Therefore, to reduce fruit postharvest decay caused by pathogens, various chemical fumigants, such as methyl bromide, sulfur dioxide, phosphine, and ethyl formate have been used to protect the fruit from pathogen infections [4]. However, fumigation using chemicals has a negative impact on the environment and these chemicals are toxic to human health. Nowadays, the research into postharvest technology focuses on applying environmentally friendly technologies to replace the use of chemicals to control pathogens and maintain the quality of fruits and vegetables. Melatonin (*N*-acetyl-5-methoxytryptamine) is widespread in plants, animals, and humans [5,6]. The function of melatonin in plants has been widely reported [7]. For instance, melatonin, as a natural antioxidant, may play important regulatory roles in various physiological activities in plants, such as fruit development [8], fruit ripening [9], delaying fruit senescence [10], disease resistance, exerting antioxidant effects, regulating growth and development, and facilitating plant adaption to stress conditions [11]. Moreover, melatonin has been applied as pre-treatment for preventing fungal infections and reducing disease symptoms in fruits [12,13,14]. Such an application increases the resistance of apples to Marssonina blot, by promoting the expression of chitinase genes, as well as regulating hydrogen peroxide and pathogenesis-related proteins [14]. Similar effects of melatonin on inhibiting pathogen infections and prolonging the shelf-life of post-harvest fruits were also observed in apples [15], strawberries [13], and citrus fruits [12]. As a safe and effective treatment, melatonin acts not only as a signaling molecule for enhancing the resistance of plants to biotic and abiotic stresses, but also as a strong free radical scavenger in many plants [5]. Recent studies have demonstrated that the function of melatonin treatment on the enhancement of fruit disease resistance could be associated with the induction of various defense-related genes, such as those encoding peroxidase (POD), lipid-transfer proteins, chitinases, β-1,3-glucanase, and pathogenesis-related (PR) proteins [16,17]. Cao et al. [15] and Hu et al. [18] reported that exogenous melatonin treatment enhanced disease resistance in apples, peaches, and kiwis during storage by continuously increasing the activity of POD, superoxide dismutase (SOD), catalase (CAT), and ascorbate peroxidase (APX). In addition, a decrease in H_2_O_2_ content has also been reported in strawberries [13], citrus fruits [12], and peaches [19]. Although the roles and functions of melatonin in the antioxidant defense mechanism have been widely studied under different stress conditions in fruits from different plant species, the detailed role of exogenous melatonin application against biotic stress from *B. cinerea* infection in strawberries during storage is still limited. The objective of the current study was to investigate the novelty effect of exogenous melatonin treatment on the antioxidant defense system and on grey mold disease in strawberries caused by *B. cinerea*. We focused our attention on strawberries because it is one of the most economically important agricultural commodities, and gray mold disease in strawberries causes severe fruit decay both before and after harvest. Moreover, this study can provide new insights into the improvement of pathogen resistance by melatonin in non-climacteric fruits during postharvest storage. In this study, we summarize the novel beneficial effects of melatonin treatment as a useful technique for improving postharvest preservation of fruit, even in non-climacteric fruits in the future.

## 2. Materials and Methods

### 2.1. Fruit Material

Strawberries (*Fragaria × ananassa* cv. Camino Real) used in the experiments were harvested from an orchard located in Ponchatoula, Louisiana, USA, at the commercial harvesting stage (approximately 80% ripened) during the spring season (March through May), and they were delivered to the laboratory at the Department of Plant Pathology and Crop Physiology, at the Louisiana State University Agricultural Center, Baton Rouge, USA, within 1 h of harvesting. The fruit were selected for uniformity and were free of any physical damage and apparent diseases.

### 2.2. Exogenous Melatonin (MT) Treatment and Botrytis cinerea Artificial Inoculation

The selected strawberry fruits were cleaned with tap water before treatment. The strawberries that were surface disinfected with 5% sodium hypochlorite (NaOCl) for 15 min were labelled as the positive control, and the negative controls were strawberries immersed in distilled water for 15 min. The strawberries immersed in 100 µM of MT for 15 min were labelled as MT-treated.

This study included the following six treatments: (1) Control (no treatment and non-inoculated); (2) Inoculated control (no treatment + *B. cinerea*); (3) Positive control (treated with 5%NaOCl); (4) Inoculated positive control (5% NaOCl + *B. cinerea*); (5) MT-treated (100 µM MT); and (6) MT-treated and inoculated (100 µM MT + *B. cinerea*).

Artificial inoculations were conducted 24 h after MT or 5% NaOCl treatment. The *Botrytis cinerea* isolate was collected from infected strawberries that showed typical symptoms of gray mold. The *Botrytis cinerea* isolate was used for inoculum production and was cultivated on potato dextrose agar (PDA) for 5 days at 25 °C under 12 h cycles of dark and blue-red lights (440–460 nm and 620–630 nm). Spores were suspended from the petri dishes with 50 mL of sterile distilled water supplemented with 0.1% (*v*/*v*) Tween 80 using a scalpel. The spore concentration of the inoculum suspension was adjusted at 2 × 10^5^ spores mL^−1^ after counting the spores using a hemocytometer. A wound-injection method by Testempasis et al. [20] with a slight modification was used to infect the strawberry fruits with *B. cinerea* to ensure consistent inoculum and to avoid the unintended destruction of the fruit. Each strawberry fruit was injected with 50 µL of spore suspension using a sterile syringe needle at a depth of about 5 mm. Following artificial inoculation, the fruits were placed in plastic trays and stored at room temperature (25 ± 2 °C) with 80–85% relative humidity for 6 days. The non-inoculated controls were injected with 50 µL of sterile distilled water supplemented with 0.1% (*v*/*v*) Tween 80, and were stored under the same conditions as the inoculated ones. On each sampling date, the treatment was replicated in triplicate, and each replication consisted of ten strawberries (30 randomly sampled fruits were taken from each treatment).

The physicochemical quality, including visual appearance, disease assessment, fruit color, weight loss, fruit firmness, total soluble solids content (TSS), and titratable acidity (TA), 2,2-diphenyl-1-picrylhydrazyl (DPPH) scavenging capacity, and antioxidant enzyme activities were determined every two days during storage. The whole fruits from the same treatment in each parameter study were pooled, frozen in liquid nitrogen, and stored at −80 °C for later analysis. Fruits were also regularly collected for chemical analysis from the replicated treatments. This study was repeated twice.

### 2.3. Disease Assessment

The severity of gray mold disease was evaluated by observing the decay area on the fruit surface. Each fruit was rated from 1 to 4 based on the following disease rating scores: 1 = no occurrence of fruit surface infection; 2 = moderate (1–35% of the fruit surface is infected); 3 = severe (36–70% of the fruit surface is infected); and 4 = very severe (71–100% of the fruit surface is infected), as shown in Figure 1. The gray mold disease index (DI) was calculated as ∑ (disease scores × number of fruit at that score)/total number of fruits in each group (30 fruits in the present experiments).

### 2.4. qPCR Quantification of B. cinerea Biomass in Strawberry Fruit

DNA was extracted from fresh fruit using a modified CTAB method, as described by Doyle and Doyle [21]. The yield and purity of the extracted DNA were determined using a NanoDrop ND-1000 spectrophotometer (Thermo Scientific, Wilmington, DE, United States). The extracted DNAs were adjusted to the same concentration (100 ng µL^−1^) and used as templates for the PCR. The sequences of specific *Bc3* primers targeting the pathogen intergenic spacer region between 28S rDNA and 18S rDNA are as follows: 5′-GCTGTAATTTCAATGTGCAGAATCC-3′ (forward) and 5′- GGAGCAACAATTAATCGCATTTC-3′ (reverse). The reaction was prepared at 1X final concentration in a 20 µL volume containing 1 µL of each primer (10 µM) and 5 µL of the diluted genomic DNA as the template. The primer pair (St-EF-forward, 5′-TGCTGTTGGAGTCATCAAGAATG-3′ and St-EF-reverse, 5′-TTGGCTGCAGACTTGGTCAC-3′ as reverse)] that amplified the elongation factor gene of the strawberry was used as an internal control to normalize the level of *B. cinerea* biomass.

### 2.5. Physicochemical Quality Measurements

The skin color of the fruit in the center of the fruit surface was measured using a chromometer (CTI color Analyzer Digital Precise Colorimeter & Color Difference Meter, China). Color measurements contained the three-parameter color system formulated by the International Commission on Illumination (CIE): L*, a* and b*. L* value represents lightness to darkness (100–0), the a * value represents redness or greenness (−greenness to +redness), and the b* value represents blueness or yellowness (−blueness to +yellowness).

Total soluble solids (TSS) content measurement was performed with fruit juice using a digital refractometer (Milwaukee’s Instruments MA871, Milwaukee Instrument, Inc., Rocky Mount, NC, USA) and the TSS data were presented as °Brix.

Titratable acidity (TA) was determined using the titrimetric method [22]. The result was expressed as g of citric acid per 100 g of fruit fresh weight.

Fresh weight loss of the strawberry fruit was measured before treatment and at different sampling days. The percentage of weight loss during storage was calculated by comparing the initial weight.

Fruit firmness was measured at two opposite sides of each individual fruit using an Effegi firmness penetrometer (model NEWTRY GY-2 Fruit Firmness Penetrometer, Milwaukee Instrument, Inc. NC, USA) with a 0.5-cm cylindrical plunger. The plunger was inserted at a depth of 0.5 cm, and the data were expressed as Newton (N). Each fruit was punctured twice in the middle part of the fruit.

### 2.6. Antioxidant Capacity Measurement

The 2,2-diphenyl-1-picrylhydrazyl (DPPH) scavenging capacity was determined according to Brand-Williams et al. [23]. For the DPPH scavenging capacity, frozen fruit powder (2 g) was homogenized with 3 mL of 85% methanol, and then the filtrated fruit extract was centrifuged at 6000× *g* for 10 min at 4 °C. A seventy µL supernatant of the strawberry fruit extract was added to a test tube and mixed with 2930 µL of 200 µM DPPH solution in methanol. The reaction mixture was incubated in the dark at room temperature for 30 min. The absorbance of the solution was read at 517 nm, and calculated using the following formula: DPPH scavenging capacity (%) = [1 − (A0/A1)] × 100. Where A0 is the absorption of the sample, and A1 is the absorption of the blank DPPH solution.

### 2.7. Antioxidant Enzyme Activity Measurement

Two grams of strawberry fruit were homogenized in 20 mL of 50 mM sodium phosphate buffer (pH 7.8) containing 1.0 mM EDTA, 0.3% (*v*/*v*) Triton X-100, and 1% (*w*/*v*) polyvinyl polypyrrolidone (PVPP) for 3 min. Following the filtration of the homogenate through cheesecloth, the filtrate was centrifuged at 15,000× *g* for 20 min at 4 °C. The supernatant liquid was used as the crude extract of antioxidant enzymes. The activities of the antioxidant enzymes were assayed using a modified method described by Suiubon et al. [24]. In order to calculate the specific enzyme activity, the protein content was measured using the method described by Bradford [25] using bovine serum albumin as a standard.

The SOD (EC 1.15.1.1) activity was assayed according to the ability to inhibit the photochemical reduction of nitro blue tetrazolium (NBT). The final assay volume was 2 mL, that consisted of 0.8 mL of distilled water, 0.5 mL of 50 mM sodium phosphate buffer (pH 5.0), 0.1 mL of 20 μM NBT dissolved in 70% ethanol, 0.2 mL of 22 μM methionine, 0.2 mL of 0.1%(*v*/*v*) Triton-X, 0.1 mL of 0.6 μM riboflavin as an enzyme substrate, and 0.1 mL of the above crude extract. The absorbance of the mixture was measured at 560 nm. One unit of SOD activity was defined as the amount of enzyme that inhibited 50% NBT at 560 nm. Total enzyme activity was expressed as units mg^−1^ protein.

The CAT (EC 1.11.1.6) activity was assayed in a 3 mL reaction volume that consisted of 0.1 mL of crude enzyme extract and 2.9 mL of 25 mM H_2_O_2_. The absorbance was measured at 240 nm every 30 s for 3 min with distilled water as the bank. One unit of CAT activity was defined as a decrease in absorbance at 240 nm of 0.01 per minute and the data were expressed as units mg^−1^ protein.

The POD (EC 1.11.1.7) activity was determined based on the ability to convert guaiacol to tetraguaiacol. The reaction mixture consisted of 1 mL of sodium phosphate buffer (50 mM, pH 5.5), 1 mL of 25 mM guaiacol, 0.5 mL of 2% (*v*/*v*) H_2_O_2,_ and 0.5 mL of crude enzyme extract. The absorbance was measured at 470 nm every 30 s for 3 min using distilled water as the bank. One unit of POD activity was defined as the amount of the enzyme that caused an absorbance change of 0.01 per minute and the data were expressed as units mg^−1^ protein.

The APX (EC 1.11.1.11) activity was measured based on the decrease in ascorbic acid. The assay reaction solution (2.0 mL final volume) contained 0.5 mL of 100 mM potassium phosphate buffer (pH 7.0), 0.5 mL of 1 mM ascorbic acid, 0.5 mL of 0.4 mM EDTA, 0.02 mL of 10 mM H_2_O_2_, 0.38 mL of distilled water, and 0.1 mL of crude enzyme extract. The absorbance of the mixture was measured at 290 nm. One unit of enzyme activity was defined as the increase of absorbance per minute and the data were expressed as units mg^−1^ protein.

### 2.8. Statistical Analysis

The experiment data are the average of three replications ± SD. An analysis of variance (ANOVA) was calculated, and the means separations were determined using Duncan’s multiple range test (DMRT) at *p* < 0.05. All experiments were repeated twice at later dates.

## 3. Results

### 3.1. Visual Appearance and Disease Development

The visual appearances of the strawberries immersed in MT solution, NaOCl, and the control with or without being inoculated with *B. cinerea* before they were stored at room temperature for 6 days, are shown in Figure 2. The incidence of gray mold disease appeared on day 2 of storage and continued to increase thereafter in all treated fruits except in the MT-treated fruit with or without artificial inoculation with *B. cinerea*. Interestingly, the symptoms of gray mold disease did not appear on the positive control (NaOCl), MT, and MT + *B. cinerea* treated fruits during the 6 days of storage at room temperature, whereas the inoculated positive control (NaOCl + *B. cinerea*) fruits were found to be infected, indicating that MT was more effective for controlling gray mold than NaOCl.

The result of the calculated gray mold disease index shown in Figure 3A reveals that the highest disease index score was found first in the inoculated control and then in the control and inoculated positive control (NaOCl + *B. cinerea*), respectively, whilst no difference was found in the disease index on the strawberry fruits immersed in the positive control (NaOCl), MT solutions, and MT + *B. cinerea* stored for 4 and 6 days. At the end of storage, the control fruits and inoculated control (+*B. cinerea*) showed visible fruit decay from *B. cinerea* infection with a disease index score of 2.0–3.0.

To validate these results, the biomass of *B. cinerea* was quantified based on the relative level of the *B. cinerea* genomic DNA compared to the strawberry DNA, as shown in Figure 3B. *B. cinerea* was detected in strawberry fruit one day after inoculation with no visual appearance of gray mold infection. The inoculated control (+*B. cinerea*) strawberry fruit had more *B. cinerea* biomass than the other treated fruits during 6 days of storage. Whereas MT-treated fruit had significantly less *B. cinerea* biomass than other treatments during storage for 4 and 6 days. On day 6 after treatment, the fungal biomass in strawberry fruit immersed in MT then inoculated with or without *B. cinerea* were reduced by 5.28- and 3.52-fold, respectively, compared to the inoculated control (+*B. cinerea*) fruit. The fungal biomass in the fruit immersed in NaOCl then inoculated with or without *B. cinerea* were reduced by 1.76- and 3.24-fold, respectively, when compared to the inoculated control (+*B. cinerea*) fruit. Moreover, the results found that there were no significant differences in *B. cinerea* biomass between the control and inoculated positive control (NaOCl + *B. cinerea*) treated fruit, whereas the *B. cinerea* biomass in MT+ *B. cinerea* treated fruit was 3.19-fold less than that in the control fruit (Figure 3B).

### 3.2. Physicochemical Qualities: Fruit Colour, TSS, TA, Fresh Weight Loss, and Fruit Firmness

The color parameters (L*, a* and b* values) of the strawberry fruit stored for 6 days were also determined (Table 1). The L* value of the strawberry fruit refers to the lightness and is normally used for evaluating the darkness of the postharvest fruit. The initial average value of L* of the strawberry fruits used in this study was 30.48, which continuously decreased in all fruit samples, indicating darkening of the fruit with time. Following 2 days of storage, MT and MT + *B. cinerea* treated fruits had the highest L* value when compared to the other treatments. However, the strawberry fruit from the control and inoculated control (+*B. cinerea*) groups had the lowest L* value and that significantly decreased after 4 and 6 days of storage (*p* < 0.05). The L* value of the positive control (NaOCl) fruit was significantly lower than that of the MT and MT+ *B. cinerea* treated fruit, and it was similar to that of the control fruit on the last day of storage. This indicates that MT was effective in maintaining the brightness of the strawberry fruit during storage, even though the fruits were infected with *B. cinerea*.

The initial average value of a* and b* of the strawberries used in this study were 22.68 and 23.67, respectively. The a* and b* values of the strawberries significantly increased during the 6 days of storage (*p* < 0.05). On day 6, the a* values of the MT and MT+ *B. cinerea* treated fruits were significantly lower than those of all other treated strawberries, indicating a delayed change in the fruit color. The b* values of the fruits from all treatments were similar.

It is widely accepted that TSS and TA of the fruits are related to the ripening process.

TSS content of the strawberries increased during the 6-day storage (Table 1). A rapid increase in TSS was found in the control and inoculated control (+*B. cinerea*) fruits, and it was significantly higher than in other treatments (*p* < 0.05), indicating that MT or NaOCl treatment delays fruit ripening. However, the TSS of the strawberries immersed in NaOCl and MT solutions with or without *B. cinerea* inoculation, increased slightly and did not differ significantly among the treatments on days 4 and 6 of storage. In this study, the initial TA of the strawberries was 1.02% and it decreased slightly during storage. No significant difference in TA was observed among the treatments within the first 4 days of storage. At the end of storage, TA of MT and MT+ *B. cinerea* treated fruits remained constant and was higher than in the other treatments.

Generally, the accelerated fruit weight losses are the result of an increased rate of transpiration and respiration. The results in Figure 4A showed that the weight loss of the strawberry fruits increased continuously during the 6 days of storage at room temperature in all treatments. The weight loss of the control and inoculated control (+*B. cinerea*) fruits increased dramatically, while that of NaOCl and MT with and without *B. cinerea* inoculation increased only slightly. At the end of storage, the highest weight loss was found in the inoculated control (+*B. cinerea*) fruit (25.25%), followed by the control, MT+ *B. cinerea,* positive control (NaOCl), inoculated positive control (NaOCl + *B. cinerea*), and MT fruits (22.60, 15.86, 13.29, 12.16, and 12.11%, respectively).

Furthermore, the fruit firmness decreased significantly for most of the treatments except that of the positive control (NaOCl) and MT solution-treated fruits, which remained fairly stable during the 6 days of storage (Figure 4B). There was no significant difference in the firmness between the control and inoculated control (+*B. cinerea*) until day 6 when a rapid decrease in fruit firmness in the inoculated control (+*B. cinerea*) fruit was observed.

### 3.3. Antioxidant Capacity and Antioxidant Enzymes

The non-enzyme antioxidant or antioxidant capacity of the strawberries during storage was expressed as the DPPH scavenging capacity. The antioxidant capacity of exogenous MT treatment was similar to the MT+ *B. cinerea* and increased significantly (*p* < 0.05) at days 4 and 6 during storage compared to other treatments (Figure 5). The antioxidant capacity of the control, inoculated control (+*B. cinerea*), positive control (NaOCl), and inoculated positive control (NaOCl+ *B. cinerea*) fruits remained constant throughout the storage period and were not found to be significantly different from one another (Figure 5).

The activities of SOD, CAT, POD, and APX were determined in strawberry fruits during the storage period and are shown in Figure 6A–D. The results reveal that SOD activities of the strawberry fruits in the control and inoculated control (+*B. cinerea*) remained constant throughout storage, while that in the positive control (NaOCl) and inoculated positive control (NaOCl + *B. cinerea*) fruits increased slightly, but less than that of the MT and MT+ *B. cinerea*. The highest level of SOD activity was detected in the MT-treated and MT-treated and inoculated fruits, which was significantly higher than that of the other treatments (*p* < 0.05) from day 2 to day 6 (Figure 6A).

The CAT activity of the strawberry fruits in all treatments fluctuated during storage. At days 4 and 6 of storage, the control fruits were found to have the lowest CAT activities while the MT treatment appeared to significantly induce the CAT activities compared to the control from days 2 to 4 of storage (*p* < 0.05) (Figure 6B). As for the POD, its activity increased in all treatments during storage (Figure 6C). Compared to the control, a significant increase in POD activity was observed at day 2 of storage in either NaOCl or MT-treated fruits with or without *B. cinerea* inoculation. This significance was maintained at day 4 and day 6 for the MT-treated group, and at day 6 for MT and MT + *B. cinerea* treated fruits. In addition, *B. cinerea* inoculation appeared to significantly reduce the POD activity (*p* < 0.05) compared to the control during the initial four days of storage (Figure 6C).

The MT treatment also significantly induced the APX activity of strawberry fruits during storage with or without *B. cinerea* inoculation when compared to the control (Figure 6D). The APX activity of strawberries treated with exogenous MT increased sharply and reached its maximum level between days 2 and 4 of storage and this was significantly higher than the other treatments (Figure 6D). Later, APX activity of MT-treated fruit slightly decreased, and no differences were found when compared with the MT +*B. cinerea* fruit at day 6 of storage (*p* < 0.05). APX activities of the control and inoculated control (+*B. cinerea*) fruits remained constant throughout in the duration of storage (Figure 6D). The NaOCl treatment also seems to increase the APX activity, however, at a slower pace and for a shorter duration.

## 4. Discussion

Exogenously applied melatonin markedly reduced disease incidence in both naturally infected and artificially *B. cinerea* inoculated strawberry fruits. It is more effective than the commonly used NaOCl method, which can only reduce natural infection in strawberry fruits during storage (Figure 2 and Figure 3). Recently, exogenous MT has been evaluated as an effective postharvest treatment to improve fruit quality and disease resistance in various fruits during storage, such as peaches [19], apples [15], strawberries [13,26], kiwis [18], and plums [27]. However, exogenously applied MT treatment had no significant effect on the postharvest green mold in citrus fruits caused by *Penicillium digitatum* [12], and anthracnose disease in banana fruits caused by *Collectotrichum musae* [17]. Moreover, Lee et al. [28] reported that Arabidopsis serotonin N-acetyltransferase (*SNAT*) knockout mutant plants exhibit a decreased MT synthesis resulting in the increased susceptibility to the pathogen, suggesting a direct involvement of MT in modulating plant disease resistance.

In the present study, strawberry fruits treated with MT exhibited a delay in fruit color development, and this has been reported in several kinds of fruits, such as bananas [10], kiwis [18], and mangoes [29]. Moreover, MT has been found to slow down the increase of TSS in bananas [10], cherries [30], and mangoes [29], which delays the ripening of climacteric fruit. These results revealed that MT could not only inhibit the ripening process of climacteric fruits, but also delay the physicochemical changes of non-climacteric fruits during storage, as shown in the strawberries in the present study. Onik et al. [31] reported that the application of MT was effective in reducing the respiration rate of apples, which resulted in slowed fruit weight loss. Therefore, it appears that MT has a high efficiency in maintaining weight loss and fruit firmness of strawberries during storage. These agree with the findings of previous studies, which showed that exogenous MT maintained the fruit firmness in bananas [10], pears [32], and mangoes [29]. Reducing weight loss and maintaining fruit firmness by exogenous MT application might be due to maintaining and improving membrane function, and reducing the activity of cell wall hydrolases resulted in delaying degradation of the cell wall [33].

Reactive oxygen species (ROS) are the early signaling molecules in plant infections by pathogens and play a role in fruit deterioration during the postharvest storage. For scavenging ROS and alleviate ROS toxicity, fruits have developed broad and efficient enzyme and non-enzyme antioxidant defense systems to induce disease resistance and protect against oxidative damage [34]. The present data show that the exogenous application of MT enhanced the antioxidant capacity in strawberry fruits infected by *B. cinerea* during storage at room temperature. In similar studies, the enhanced antioxidant capacity by exogenously applied MT was reported in strawberry fruits during storage at 4 °C [13,26], in pomegranates [35], and mangoes [36].

It is widely recognized that the enzymatic antioxidant system is an important regulator of ROS in plant cells. SOD, CAT, POD, and APX are the key antioxidant enzymes for scavenging ROS. Plants produce a series of ROS under stress. It is widely recognized that SOD removes O^2−^ whilst CAT, POD, and APX remove H_2_O_2_ to prevent oxidative damage to plant cells [37]. In our experiments, MT consistently induced the SOD, POD, and APX activities in strawberries during storage and also against the damage induced by *B. cinerea* infection. However, MT did not seem to affect the CAT activity of strawberries during storage, which did not vary notably among different treatments. This is unlike the results with peaches and kiwis, where MT treatment induced the activities of CAT, POD, and APX during storage at the postharvest senescence stage [18,19].

These results demonstrate that exogenous MT could be used to improve plant disease resistance to reduce *B. cinerea* infection by upregulating antioxidant defense systems. This is in agreement with the study by Cao et al. [15] who found that the MT treatment reduced pathogen infection in apples by continuously increasing the activities of antioxidant enzymes. MT upregulated the activity of antioxidant enzymes in peaches and reduced the levels of O^2−^ and H_2_O_2_ in different peach varieties, therefore maintaining the metabolism balance of ROS and delaying fruit deterioration [19]. Our data also indicated that MT is able to mobilize the defense-related antioxidant enzymes to scavenge the produced ROS through the increase the enzymatic activities of SOD, POD, and APX enzymes to at least partially convert H_2_O_2_ to H_2_O, resulting in reduced toxicity to plant cells.

## 5. Conclusions

The current study demonstrated that exogenously applied MT is an effective alternative to NaOCl treatment in reducing the symptoms of gray mold disease caused by *B. cinerea* in strawberries during storage at room temperature. This is likely due to the higher induction of activities of the enzymatic antioxidants, such as SOD, POD, and APX, to detoxify the reactive oxygen compounds produced during storage. In addition, in the case of strawberries (non-climacteric fruit), MT treatment maintained the fruit quality based on visual appearance, fruit color, TSS, TA, fresh weight, and fruit firmness. This study provides a better alternative approach for the post-harvest handling of strawberries to extend their shelf life during storage at room temperature.

## Figures and Tables

**Figure 1 foods-12-01445-f001:**
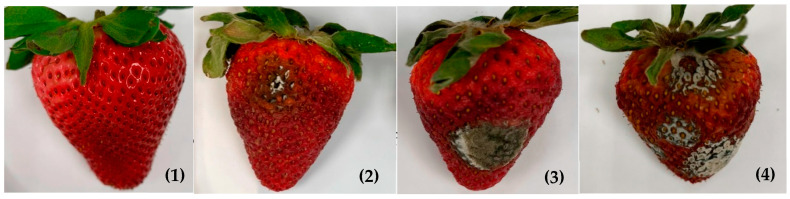
Gray mold disease rating scores (**1**–**4**) of strawberry fruit.

**Figure 2 foods-12-01445-f002:**
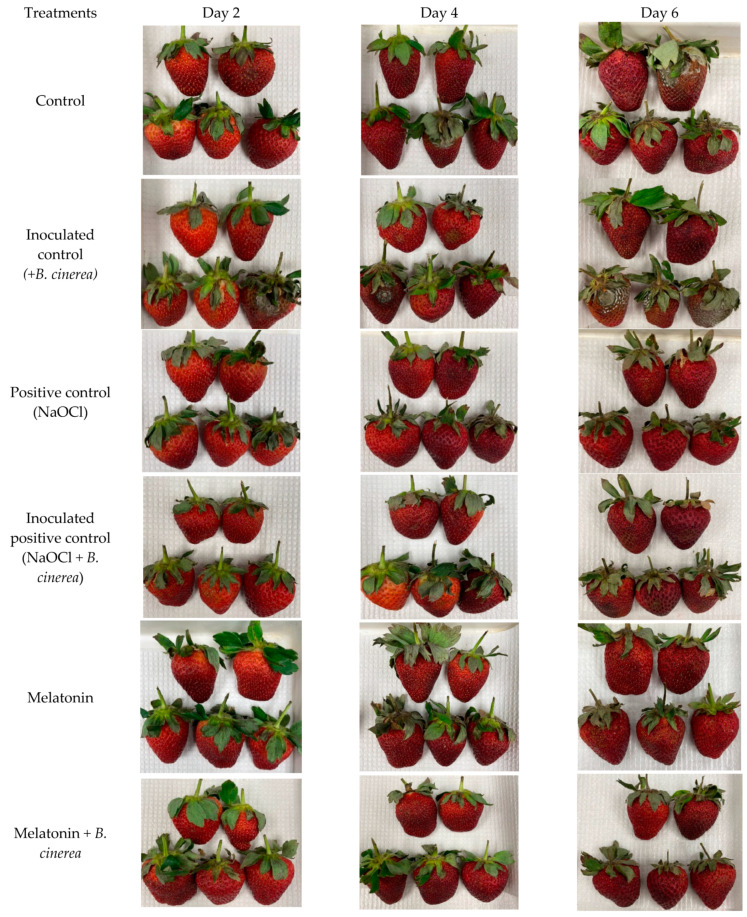
Effect of exogenous melatonin (MT) on the visual appearance of strawberry fruits inoculated with and without *Botrytis cinerea,* and stored at room temperature (25 ± 2 °C, 80–85% RH) for 6 days.

**Figure 3 foods-12-01445-f003:**
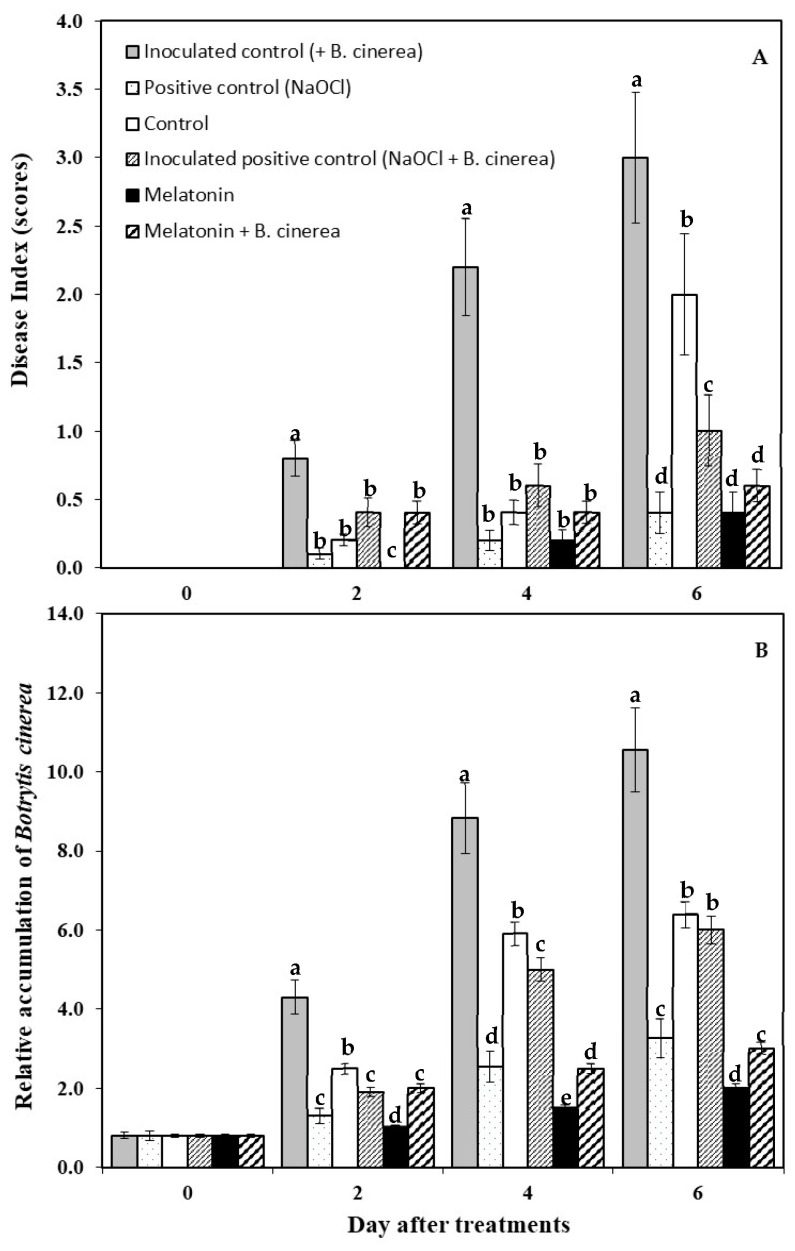
Disease index (**A**) and biomass of *Botrytis cinerea* were measured using real time PCR (**B**) in strawberry fruits treated with exogenous melatonin and inoculated with and without *Botrytis cinerea*, and stored at room temperature (25 ± 2 °C, 80–85% RH) for 6 days. Values are the average of three replicates ± SD. Different letters indicate significant differences according to Duncan’s multiple range test (*p* < 0.05).

**Figure 4 foods-12-01445-f004:**
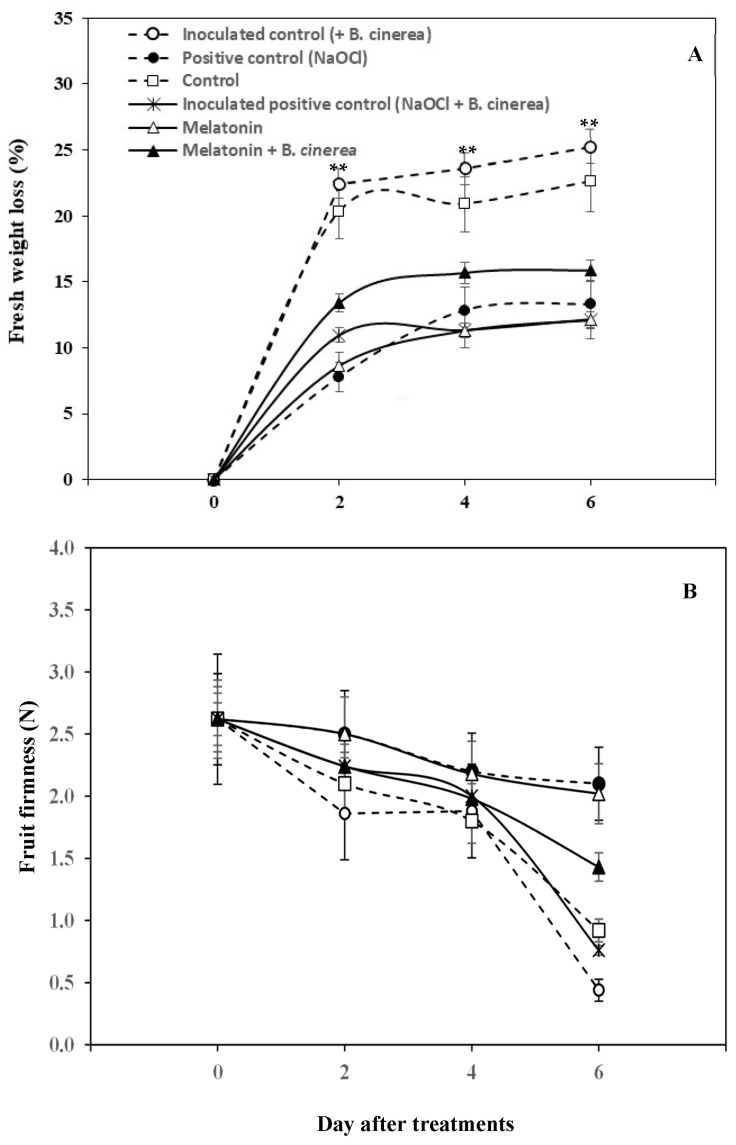
Fresh weight loss (**A**) and fruit firmness (**B**) of the strawberries treated with exogenous melatonin and inoculated with and without *Botrytis cinerea* during storage at room temperature (25 ± 2 °C, 80–85% RH) for 6 days. Data are the means ± S.D. of three replications (three sets of ten fruits). Significant differences between treatments are indicated with asterisks (** (*p* < 0.01)).

**Figure 5 foods-12-01445-f005:**
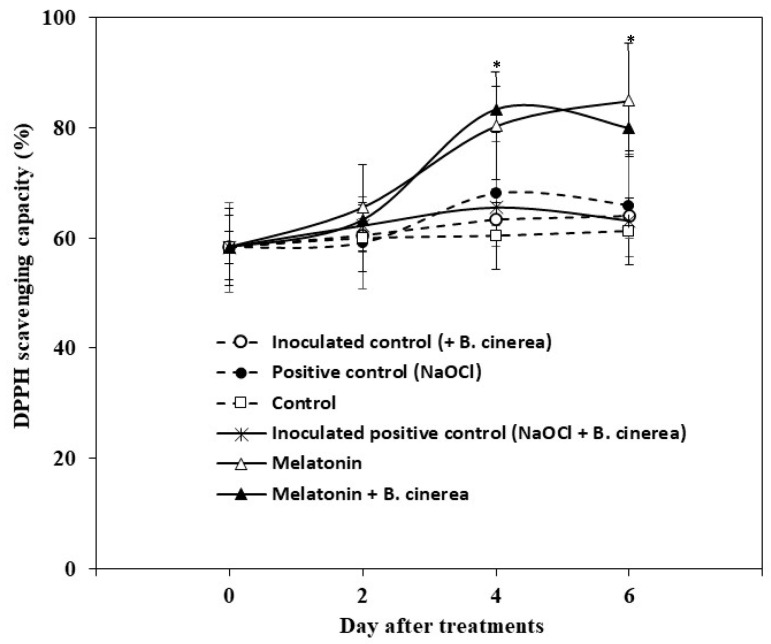
DPPH scavenging capacity of the strawberries treated with exogenous melatonin and inoculated with and without *Botrytis cinerea* during storage at room temperature (25 ± 2 °C, 80–85% RH) for 6 days. Data are the means ± S.D. of three replications (three sets of ten fruits). Significant differences between treatments are indicated with asterisks (* (*p* < 0.05)).

**Figure 6 foods-12-01445-f006:**
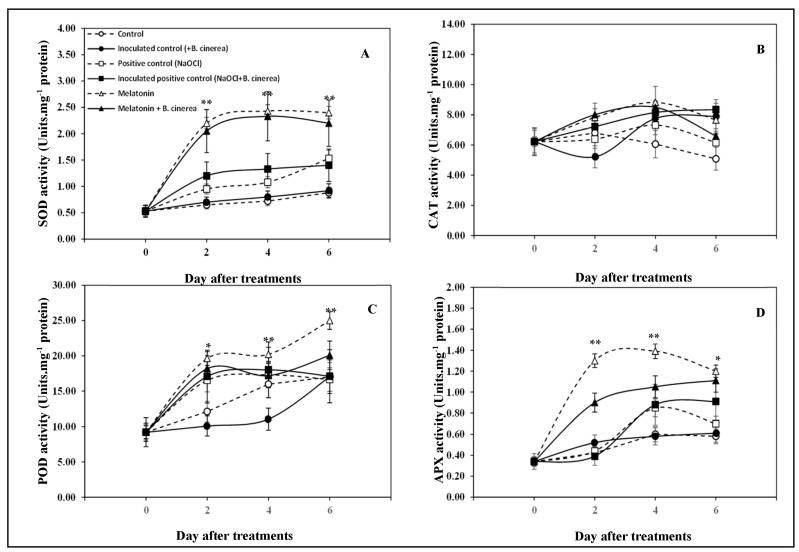
SOD (**A**), CAT (**B**), POD (**C**), and APX (**D**) activity of the strawberry fruits treated with exogenous melatonin and inoculated with and without *Botrytis cinerea* during storage at room temperature (25 ± 2 °C, 80–85% RH) for 6 days. Data are the means ± S.D. of three replications (three sets of ten fruits). Significant differences between treatments are indicated with asterisks (** (*p* < 0.01); * (*p* < 0.05)).

**Table 1 foods-12-01445-t001:** Effects of melatonin on fruit color and other physicochemical qualities (total soluble solids, TSS; total titratable acidity, TA) of strawberry fruits treated with exogenous melatonin, and inoculated with or without *Botrytis cinerea* during a six-day storage at room temperature (25 ± 2 °C, 80–85% RH).

Day after Treatment	Treatments	Color ^§^			TSS (°Brix)	TA (% Citric Acid)
		L*	a*	b*		
Day 0		30.48 a	22.68 c	23.67 b	6.14 c	1.02 a
Day 2	Inoculated control (+*B. cinerea*)	24.69 c	26.32 a	23.99 b	6.02 c	0.72 b
	Positive control (NaOCl)	25.89 c	26.96 a	24.60 b	6.32 c	0.70 b
	Control	25.53 c	26.22 a	25.35 a	6.56 c	0.65 b
	Inoculated positive control (NaOCl *+ B. cinerea*)	24.49 c	26.48 a	25.29 a	6.82 c	0.63 b
	Melatonin	27.16 b	23.53 b	24.30 b	6.54 c	0.78 b
	Melatonin + *B. cinerea*	26.19 b	23.24 b	24.00 b	6.22 c	0.71 b
Day 4	Inoculated control (+*B. cinerea*)	22.30 d	26.91 a	23.95 b	8.00 a	0.68 b
	Positive control (NaOCl)	24.70 c	26.75 a	24.49 b	7.42 b	0.66 b
	Control	23.55 d	27.21 a	25.53 a	8.10 a	0.60 b
	Inoculated positive control (NaOCl *+ B. cinerea*)	24.45 c	26.44 a	25.37 a	7.86 b	0.59 b
	Melatonin	26.55 b	24.56 b	24.70 b	7.64 b	0.64 b
	Melatonin + *B. cinerea*	26.96 b	24.79 b	24.50 b	7.38 b	0.68 b
Day 6	Inoculated control (+*B. cinerea*)	19.74 e	28.54 a	25.15 a	8.30 a	0.50 c
	Positive control (NaOCl)	23.19 d	26.65 a	25.64 a	7.46 b	0.51 c
	Control	22.09 d	28.67 a	26.57 a	8.22 a	0.55 c
	Inoculated positive control (NaOCl *+ B. cinerea*)	22.24 d	28.54 a	25.77 a	7.76 b	0.53 c
	Melatonin	25.55 c	25.36 b	25.35 a	7.32 b	0.61 b
	Melatonin + *B. cinerea*	25.65 c	25.74 b	26.59 a	7.98 b	0.68 b

Data are the means of three biological replications, each consisting of ten fruits. ^§^ Within each column, a value followed with the same letter does not differ significantly at *p* < 0.05.

## Data Availability

Data is contained within the article.

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
