# Peer review of "Melatonin Treatment of Strawberry Fruit during Storage Extends Its Post-Harvest Quality and Reduces Infection Caused by Botrytis cinerea"

_foods, 2023, doi:10.3390/foods12071445_

Round 1
Reviewer 1 Report
This paper deals with the effects of exogenous melatonin on the physicochemical quality, antioxidant defense system, and disease resistance of strawberries to Botrytis cinerea infection.
The results show that the treatment effectively maintains the strawberry's postharvest quality even in inoculated fruits.
This interesting and practically applicable work is within the Aims and Scope of the journal.
The article is concise and clearly written, implementing all standards of scientific work.
It would be interesting to continue the research on the melatonin effects on other berry fruits.
I suggest following corrections:
Line 37: check the value of 4,510 tons (is it too small?)
53-54: Arnao and Hernández- Ruiz (2006 or 2015?)
89-95: specify the season in which the strawberries were produced
152: check the reference year in the text and in the References
324-325: check 3.38 and 8.8-fold; the same for lines 327, 330 (I do not understand how the authors calculated these numbers, it does not match with Figure 3B)
337-338: there are no letters in the figure
357-358: delete the first sentence, it is already in the table heading
483-485: rearrange for clarity
504: mango
In my opinion, there is no need for other corrections,
Reviewer 2 Report
Dear Editor, in the manuscript Foods-2300146 authors evaluated effects of exogenous melatonin (MT) treatment on physicochemical quality traits, antioxidant defence system and disease resistance of strawberry fruit to Botrytis cinerea infection. Results showed that 100 μM MT dipping treatment for 15 min maintained fruit colour, titratable acidity and firmness when compared to the untreated control fruit and those treated with 5% NaOCl. In addition, MT increased the accumulation of DPPH scavenging capacity and the activity of antioxidant enzymes (SOD, POD and APX) with the exception of CAT. The same effect was also observed in strawberry fruit after being immersed in MT and followed by B. cinerea inoculation.
The effects of melatonin treatment on strawberry quality properties, antioxidant system and fungal decay have been reported in a wide range of previous papers, such as Aghdam and Fard, 2017, Food Chemistry, 221, 1650-1657; Liu et al., 2018, Postharvest Biology and Technology, 139, 47-55; El-Mogy et al., 2019, 2019, Journal of Berry Research, 9(2), 297-307; Huang et al., 2021, Shipin Kexue/Food Science, 42, 187-193; Manafi et al., 2022, Journal of Plant Growth Regulation, 41, 52-64; Hayat et al., 2022, Horticulturae, 8(3),194. The last 4 papers have not been considered in the present manuscript.
Other concerns are listed below:
- Line 40: This reference should be provided in reference list as NASS, 2021 instead of USDA, 2021.
- Use h for hour abbreviation instead of hr along the manuscript.
- Line 73: It should be Hu and Rao, according to reference list.
- Change “in different plants” to “in fruit from different plant species”
- Line 121: If you have only 10 fruits for each treatment and replicate, how many fruits were taken from each treatment and sampling date? This issue is not clear.
- Line 126: Do you mean fruit from the same replicate, treatment and sampling date? Was the whole fruit used or the decayed portion was discarded?
- Line 235: once or twice? According to information in line 128, the experiment was replicate twice.
- Figures 3, 4, 5 and 6: Significant differences among treatments for each sampling date should be shown by adding different letters.
- Line 357: Delete this sentence because this information is also provided in table head title.
- Line 412-417: This information should be moved to Discussion section.
- Lines 430-431: he same as above.
- Subtitles are not necessary in Discussion section.
- Line 502: It should be MT instead of MA.
- English grammar and spelling should be checked along the manuscript.
- References should be written according to the journal format.
Reviewer 3 Report
The study is aimed to investigate the effects of exogenous melatonin on the physicochemical quality, antioxidant defense system and disease resistance of strawberry fruit to Botrytis cinerea infection.
L12 and L32: Give the full latin name of the straberry species.
L15 and L83: B. cinerea
L28-30: delete empty rows.
L164-165: delete empty rows.
L333: The quality of figure 3 need to be improved.
L471: Figures 2 and 3
L504: mango
L551: Please format the References section according to journal format
L553: Fragaria x ananassa and in italic
L646: Botrytis cinerea - in italic
Round 2
Reviewer 2 Report
The novelty of the present manuscript should be highlighted at the end of the introduction section.
